# From Reasoning to Learning: A Survey on Hypothesis Discovery and Rule Learning with Large Language Models

**Kaiyu He**                                                        *kaiyu.he@utdallas.edu*
*Department of Computer Science*
*University of Texas at Dallas*

**Zhiyu Chen**                                                   *Zhiyu.Chen2@utdallas.edu*
*Department of Computer Science*
*University of Texas at Dallas*

**Reviewed on OpenReview:** *https: // openreview. net/ forum? id= d7W38UzUg0*

## Abstract

Since the advent of Large Language Models (LLMs), efforts have largely focused on improving their instruction-following and deductive reasoning abilities, leaving open the question of whether these models can truly discover new knowledge. In pursuit of artificial general intelligence (AGI), there is a growing need for models that not only execute commands or retrieve information but also learn, reason, and generate new knowledge by formulating novel hypotheses and theories that deepen our understanding of the world. Guided by Peirce's framework of abduction, deduction, and induction, this survey offers a structured lens to examine LLM-based hypothesis discovery. We synthesize existing work in hypothesis generation, application, and validation, identifying both key achievements and critical gaps. By unifying these threads, we illuminate how LLMs might evolve from mere "information executors" into engines of genuine innovation, potentially transforming research, science, and real-world problem solving.

## 1 Introduction

One major pillar of human intelligence is the capacity to discover hypotheses and learning rules. We call this capability *hypothesis discovery* (or rule learning). Earlier AI systems struggled with it because formal symbolic methods lacked the commonsense background needed for inventive rule formation (Yu et al., 2024a). Recent advances in natural language processing (NLP) have produced LLMs pretrained on extensive text corpora that embed substantial commonsense knowledge. These models now enable tasks that demand rich background knowledge, such as formulating new hypotheses and deriving novel conclusions.

Hypothesis discovery inherently relies on a blend of reasoning that includes abduction, induction, and deduction, each defined differently by various scholars. For instance, Gilbert H. Harman considers induction to be a special case of abduction, describing it as "inference to the best explanation" (IBE) (Harman, 1965; Douven, 2021). However, while this definition is easy to understand, it oversimplifies key aspects of hypothesis discovery. In particular, the notion of the "best" explanation is ambiguous and often requires additional assumptions that vary by context. Moreover, this framework does not fully capture real-world scenarios, where a "best" explanation is rarely reached immediately; rather, we continually experiment, gather new observations, and refine our hypotheses. Based on these considerations, we adopt Charles Peirce's definition of hypothesis discovery and reasoning, which posits that hypothesis discovery begins with forming an explanatory hypothesis to explain observations through **abduction**, proceeds with iteratively apply hypothesis to solve problem or derive new knowledge with **deduction**, and validate hypothesis through **induction** (Frankfurt, 1958; Peirce, 1974; Burks, 1946; Minnameier, 2004) (See explanation in Figure 2).

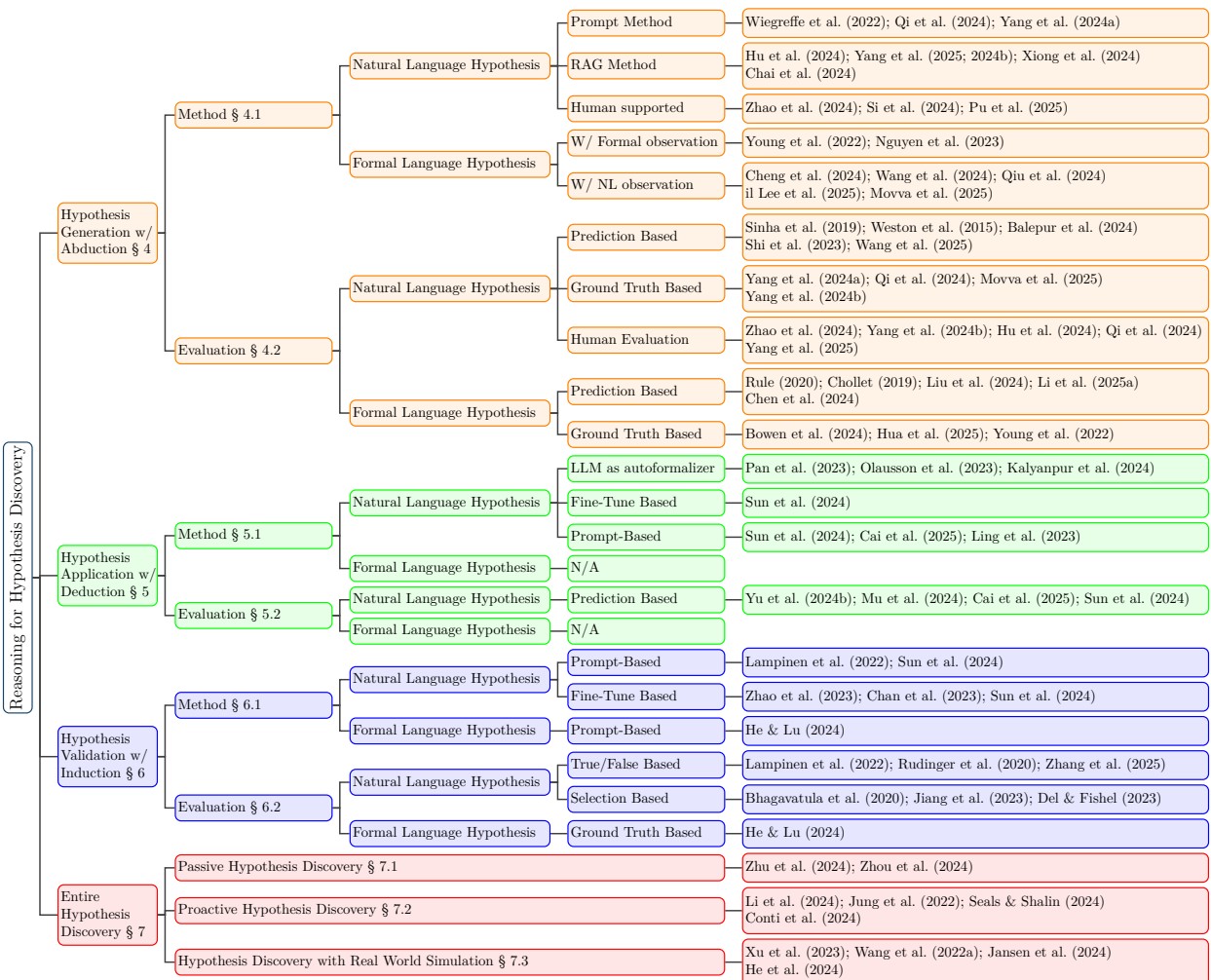

Figure 1: Taxonomy for Hypothesis Discovery with LLMs. Our survey categorizes work into four topics based on Peirce's definition of hypothesis discovery: Generation (creating hypotheses that explain given observations with abduction), Application (deducing new observations from established hypotheses with deduction), Validation (verifying and refining hypotheses against new evidence with induction), and Integrated Hypothesis Discovery (examining the dynamic interdependencies among these components in a continuous, iterative process).

The rest of the survey is organized as follows. Section 2 presents background knowledge on hypothesis discovery using LLMs, including different forms of reasoning and representations involved in the process. Section 3 examines prior surveys on LLM reasoning and hypothesis discovery, highlighting their narrow emphasis on deductive tasks or application-specific methods. Section 4 reviews methods for forming hypotheses *(Abduction)*. Section 5 then covers approaches for applying these hypotheses *(Deduction)*, and Section 6 focuses on techniques for validating given hypotheses with new observations *(Induction)*. Finally, Section 7 explores the entire hypothesis-discovery cycle by examining the interdependencies among these reasoning steps and showing how abduction, deduction, and induction can be iteratively used to refine more robust hypotheses. For each stage, we discuss methods, benchmarks, evaluations, and identify limitations and future directions. A high-level taxonomy guiding this survey is shown in Figure 1.

## 2    Background

Before LLMs, most AI systems stored knowledge as handcrafted symbols and rules. This paradigm originated in the 1960s, in which the DENDRAL (Lindsay et al., 1993) system pioneered expert deductive reasoning in chemistry but required scientists to manually encode their knowledge into computer-readable rules. Following

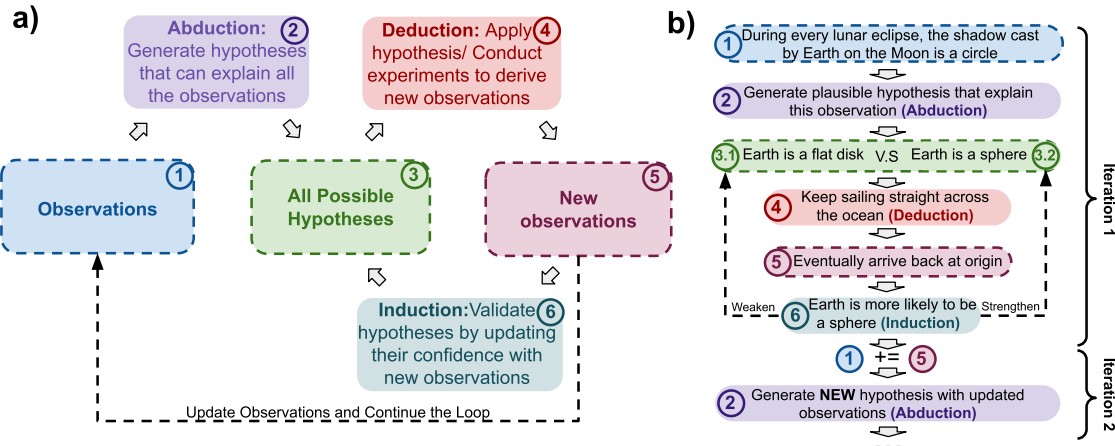

Figure 2: On the left-hand side, **a)** illustrates Peirce's framework for hypothesis discovery through abduction, deduction, and induction. The process begins with abduction, which generates explanatory hypotheses based on an initial set of observations. Deduction is then used to apply these hypotheses and derive predictions. Induction evaluates how well the predicted observations align with actual outcomes, updating the confidence of the hypotheses or rejecting those that are no longer valid. This process is iterative: validated hypotheses may be refined through further rounds of abduction using updated observations, gradually leading to more robust theories. On the right-hand side, **b)** provides a simple example that illustrates this process.

programs like BACON (Langley, 1977) and Inductive Logic Programming (Muggleton, 1991) were limited and obstacled by this fundamental knowledge representation problem. That format works well for deduction, because most of the problems we need to solve with symbolic AI systems work with limited premises and countable task-specific knowledge; even in the most recent work, for example, questions in the ProofWriter (Tafjord et al., 2021) and FOLIO (Han et al., 2024) benchmarks are limited to fewer than a hundred premises. However, abduction and induction are different: they call for generating and validating many tentative explanations inspired by vast commonsense or expert domain knowledge (such as weather patterns, social norms, or physics) and for updating beliefs as new observations arrive. Handling these reasoning tasks with symbolic AI meant writing and maintaining vast, interlocking rule bases, an effort so costly that few projects moved beyond toy domains (Yang et al., 2024c). Consequently, the research landscape remained dominated by deductive tasks (Yu et al., 2024a; Liu et al., 2025; Huang & Chang, 2023).

Later in the 2000s, when probabilistic models took charge, the Bayesian model seemed a better solution compared with symbolic AI (Pearl, 2009). However, due to the limitations of computational power, model structure, and the training data we use, small-scale probabilistic models relied heavily on human-abstracted clean features to build models. Without the main step of feature engineering by humans, automated hypothesis discovery was impossible with early probabilistic models. LLMs have transformed this landscape. Advanced model structure with billions of parameters trained on vast corpora, they implicitly absorb broad commonsense and domain knowledge, exhibiting strong reasoning abilities on complex, natural-language tasks (Yang et al., 2024c). With a simple text prompt, we can now ask them to carry out abduction or induction and even inspect their intermediate reasoning steps (Li et al., 2024; Jung et al., 2022), exposing the latent information they rely on. This advancement has made it practical to study and deploy **defeasible reasoning** (Yang et al., 2024c). Defeasible reasoning refers to forms of reasoning, such as abduction and induction, that yield probable conclusions that remain open to revision as new evidence emerges. This shift has fueled a wave of NLP research that places such flexible reasoning at the heart of AI progress (Liu et al., 2025; Huang & Chang, 2023).

## 2.1 Hypothesis Discovery

Throughout this survey, we treat *hypotheses*, *theories*, and *rules* as compressed, human-learned knowledge for understanding and predicting the world. A hypothesis is a tentative claim that may, with sufficient empirical

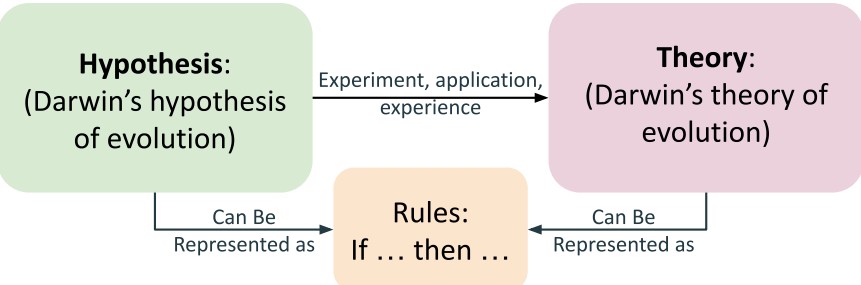

Figure 3: Relationship between hypothesis, theory, and rule.

support, mature into a theory—a more firmly established hypothesis. Both hypotheses and theories can often be cast as rules, typically written in the form "if $A$ then $B$" (See in Figure 3).

This process of Hypothesis discovery or Rule learning, the cyclical process of formulating hypotheses, gathering evidence, validating or refuting them, and ultimately establishing robust theories, lies at the heart of scientific progress (Eger et al., 2025). Early humans, for example, hypothesized that the Earth was flat based on everyday observations. Later, Eratosthenes measured shadow angles at different locations, obtaining evidence that suggested the Earth's surface was curved. This evidence challenged the flat Earth hypothesis, and subsequent findings, notably Magellan's circumnavigation, conclusively confirmed the Earth's roundness. Even with today's sophisticated instruments, researchers continue to iterate this loop in new domains, validating and refining theories as new data emerges.

Today, there is growing interest in whether LLMs can autonomously generate, apply, and validate hypotheses from natural language represented observations, mirroring this iterative process to achieve interpretable and adaptive hypothesis discovery. Although many studies have explored individual steps of hypothesis discovery, their efforts tend to be scattered across abduction, deduction, and induction, with insufficient attention to how these forms of reasoning interconnect to drive genuinely iterative, hypothesis-driven discovery.

## 2.2 Reasoning

Reasoning is central to hypothesis discovery. Researchers have historically debated how best to categorize reasoning into clear, operational types. Different frameworks each have strengths and limitations (Harman, 1965; Douven, 2021; Bacon, 1878; Laudan, 1971; Mill, 2024; Stadler, 2011; Popper, 2005; Okoli, 2023). In this survey, we adopt Charles Peirce's definition of reasoning (Peirce, 1974; Burks, 1946; Minnameier, 2004), emphasizing abduction, deduction, and induction as separate but interrelated processes. We choose Peirce's framework for three main reasons. First, clarity: Unlike many other approaches, Peirce explicitly differentiates among the three reasoning types, preventing confusion, such as the common conflation of abduction and induction. Second, practicality: Peirce's model aligns each form of reasoning directly with a distinct phase in the hypothesis discovery cycle—abduction for generating hypotheses, deduction for applying these hypotheses, and induction for validating them. This clear mapping makes his framework particularly suitable for systematically studying the entire process of hypothesis discovery, rather than isolated reasoning components. Finally, granularity: Peirce's framework breaks down the scientific discovery process into well-defined, finer-grained steps, facilitating detailed analysis and enabling more structured evaluation.

**Abductive Reasoning** is the process of forming explanatory hypotheses to make sense of observed phenomena. It is the only form of reasoning that generates entirely new ideas or explanations (Peirce, 1974; Frankfurt, 1958). Given a set of observations, one uses creative thinking and recalls necessary knowledge to come up with hypotheses that plausibly explain these observations. Importantly, a single set of observations can lead to multiple possible explanations. For instance, if you come home and find the floor wet, you might form several possible explanations: perhaps a pipe leaked, or someone spilled water accidentally. Without additional evidence or testing, you can't know for sure which explanation is correct. This illustrates how abduction helps generate potential explanations, which then must be tested further.

**Inductive Reasoning** is the process of testing whether the hypothesis and deduced consequences really obtain and evaluating to what extent they obtain (Minnameier, 2004; Peirce, 1974). In practice, induction updates a hypothesis's confidence based on new observations, including rejecting it outright, or selects the most convincing candidate from a set of competing hypotheses. Consider the claim "Swans are *(100%)* white," or linguistically, "All swans are white," formed after observing 99 white swans in Texas. Encountering a black swan in New York contradicts that hypothesis. Through induction, we recognize this contradiction and lower our confidence in the original claim, adjusting it to "Swans are *(99%)* white," linguistically expressed as "Almost all swans are white." In this example, although the hypothesis appears "revised," the change is limited to its confidence level; no new explanatory perspective is introduced, and we do not actually form a new hypothesis. By contrast, abduction can lead us to a fresh explanatory hypothesis with new observations, e.g., "Swans' color depends on their habitat," or "All swans in Texas are white," which introduces new ideas and is not a case of induction. Thus, inductive reasoning verifies or refines existing hypotheses (in terms of confidence) based on accumulating evidence.

**Deductive Reasoning** is the process of logically deriving specific conclusions from general hypotheses or rules. If the initial hypotheses are true, deduction guarantees that the derived conclusions must also be true. For instance, from the general rule "All swans are white" and the observation "This bird is a swan," we logically conclude "This bird must be white". While traditional deductive reasoning tasks, such as instruction-following and standard problem-solving, have been extensively studied with LLMs (Pan et al., 2023; Wei et al., 2022; Liu et al., 2025; Huang & Chang, 2023), deductive reasoning in the context of hypothesis discovery poses unique challenges. Specifically, it emphasizes inferential rule-following, requiring models to consistently apply hypotheses or rules to derive new and potentially unfamiliar conclusions, even when these hypotheses are counterfactual, unfamiliar, or incorrect. For example, when a flawed hypothesis is introduced in an unfamiliar domain, inferential rule-following requires us to strictly derive its predicted consequence, even if that consequence itself is incorrect. By comparing this consequence with experimental data, we can directly assess the hypothesis's validity and guide its revision. Conversely, if the deductive process is unreliable, we may overlook real contradictions and thus retain invalid hypotheses or discard valid ones. Indeed, recent work shows that although LLMs can demonstrate strong deductive performance on in-distribution tasks, they rely heavily on surface-level pattern matching and fail to generalize their inferential rule-following to novel or counterfactual scenarios (Pu et al., 2025; Mirzadeh et al., 2024; Kang et al., 2024; Yan et al., 2025b).

There are also other types of reasoning, such as analogical reasoning (Yuan et al., 2023; Jiayang et al., 2023). However, their function in hypothesis discovery is generally covered by abduction and induction. We will include these additional forms when we encounter a relevant case in the following section.

## 2.3   Rule Representation: Formal Language vs Natural Language

Table 1: Comparison of Natural vs. Formal Language Representations for the Hypothesis "Sam is a dragon". In natural language, commonsense knowledge is implicitly embedded, and derived knowledge relies on extensive commonsense, potentially resulting in different interpretations depending on background knowledge and context. In formal languages (e.g., first-order logic (FOL) or code), the knowledge base must be defined explicitly and cannot fully capture all commonsense knowledge, however, the derived conclusions are deterministic and precise.

| Representation | Hypothesis | Knowledge Base | Derived Knowledge |
|---|---|---|---|
| Natural Language | English: 'Sam is a dragon" | English commonsense of 'dragon" | Sam is dangerous |
| | Chinese: 'Sam is a dragon" | Chinese commonsense of 'dragon" | Sam brings good fortune and a bountiful harvest |
| Formal Language | FOL: Dragon(Sam) | $\forall x(\text{Dragon}(x) \rightarrow \text{Fly}(x))$ ... | Fly(Sam) |
| | Code: `Sam = Dragon()` | `Class Dragon:`
`  def fly(self):`
`    ...` | `Sam.fly()` |

There are many ways to represent hypotheses and rules, which we broadly divide into two categories: **formal languages (FL)** and **natural languages (NL)**. Formal languages, such as first-order logic and programming languages, are systematic and rule-bound. After real-world entities are encoded as explicit literals, precise inference rules yield provably correct and sound conclusions, making these systems well suited to **deductive** reasoning. Yet the encoding process strips away many subtle semantic relationships and commonsense knowledge, limiting the system's ability to handle the creative, defeasible reasoning required for **abduction** and **induction** (McCarthy & Hayes, 1981; Reiter, 1980; Hanks & McDermott, 1987; Liu et al., 2025; Yu et al., 2024a; Huang & Chang, 2023). Natural language preserves those nuances and aligns more closely with human cognition, so it is better suited to **abductive** and **inductive** tasks. However, its meanings are implicit and context-dependent, making it difficult to define a deterministic reasoning pipeline and reducing the reliability of the resulting inferences (See in Table 1). Accordingly, the following sections treat formal-language and natural-language approaches separately, emphasizing how their reasoning methods and evaluation protocols differ.

## 3 Related Surveys

Most existing work assessing LLM reasoning, both survey syntheses and popular benchmarks such as GSM8K(Cobbe et al., 2021), centres almost exclusively on multi-step deductive tasks, leaving abduction and induction, the engines of hypothesis discovery, largely unexplored. Surveys of the field Yu et al. (2024a); Liu et al. (2025); Huang & Chang (2023) highlight the absence of systematic study and clear analytical frameworks for these modes, while benchmark analyses likewise show that abductive and inductive inference receive limited attention (Plaat et al., 2024; Li et al., 2025b). This imbalance obscures our understanding of whether, and to what extent, LLMs can perform the creative, evidence-based reasoning required for hypothesis-driven discovery.

On the other hand, research in the AI for Science domain takes a distinctly **horizontal, application-driven** approach. This body of work emphasizes practical tasks such as generating research ideas, conducting experiments, and synthesizing reports, often employing domain-specific pipelines tailored to individual scientific fields. However, these studies usually lack a generalizable reasoning framework applicable across different scientific contexts. Furthermore, their evaluation metrics, typically novelty, creativity, or consistency, tend to be subjective, human-centric, and thus difficult to generalize, offering limited theoretical insight into the underlying reasoning mechanisms involved in scientific discovery (Movva et al., 2025; Alkan et al., 2025; Reddy & Shojaee, 2025; Gridach et al., 2025; Bazgir et al., 2025).

Our survey adopts a **vertical, reasoning-centered perspective** grounded in Peirce's classical framework. It integrates three modes of reasoning into a unified view of hypothesis discovery: **abduction** for hypothesis generation, **deduction** for hypothesis application, and **induction** for hypothesis validation. Unlike prior surveys that emphasize primarily deductive tasks, we concentrate on the entire reasoning process involved in hypothesis discovery, explicitly covering both defeasible reasoning (abduction and induction) and deductive reasoning. By clearly defining each reasoning mode and explaining its role within each stage of the discovery process, we provide a structured basis for designing principled, model-agnostic benchmarks and evaluation tasks. Compared to existing application-oriented surveys, our framework thus offers a more abstract, systematic, and theoretically informed approach to understanding and enhancing the role of LLMs in automated scientific discovery.

## 4 Hypothesis Generation with Abduction

Every scientific discovery begins with a set of observations, denoted as $\mathbb{O} = \{o_1, o_2, \ldots, o_n\}$, that we aim to explain. Let $h$ represent the generated explanation or hypothesis. The hypothesis generation task can be defined as generating an $h$ such that:

$$h \models (o_1 \wedge o_2 \wedge \cdots \wedge o_n)$$

This notation means that $h$ logically entails the observations. In other words, assuming $h$ holds, it guarantees that all observations $o_1 \wedge o_2 \wedge \cdots \wedge o_n$ follow. In this survey, we follow Peirce's definitions for reasoning.

Accordingly, the primary process used in hypothesis generation is abduction, the method of formulating explanatory hypotheses to account for observed phenomena.

## 4.1 Method

Despite LLMs' demonstrated prowess in tasks like summarization or code generation, devising robust methods to guide them in hypothesis generation remains an active area of research. Recent work has sought to leverage LLMs' in-context learning and natural language understanding to produce novel or domain-specific hypotheses, spurring the development of new techniques aimed at improving both the quality and applicability of generated hypotheses (Yang et al., 2024c). In this section, we review these methods, spanning approaches that rely solely on prompting, those that integrate external knowledge sources, and those that incorporate human expertise in the loop.

### 4.1.1 Natural Language Hypothesis Generation with LLMs

**Prompt-Based Methods**: Due to the lack of large-scale, domain-specific data for hypothesis generation, most abduction approaches rely on prompt-based methods that are easy to deploy and don't require extensive additional data. For instance, when provided with observations expressed in natural language and asked to generate a plausible hypothesis that explains them, both Wiegreffe et al. (2022) and Qi et al. (2024) employ few-shot prompting to guide LLMs in generating hypotheses. Specifically, Wiegreffe et al. (2022) constructs few-shot examples using a triplet format *(question, answer, explanation)*. In solving a task of generating biomedical hypotheses with given observations, Qi et al. (2024) embeds a small set of independent observation-to-hypothesis pairs in the prompt. By showing how each block of biomedical background observations maps to its corresponding hypothesis, the model learns to extract relevant domain cues and generate novel biomedical hypotheses. Their findings indicate that including more examples in the prompt tends to reduce the novelty of the generated hypotheses while increasing their correctness. Furthermore, Yang et al. (2024a) propose a pipeline for hypothesis generation that involves five prompt-based modules: one to generate hypotheses, one to test deductive consistency, one to verify that the hypothesis is not merely a copy of the given context, one to assess its generalizability, and one to determine whether the hypothesis is trivial.

**RAG-Based Methods**: Labeling massive corpora for pre-training is costly, but assembling a small or medium dataset for Retrieval-Augmented Generation (RAG) is practical, and several studies follow a similar iterative three-step pattern: (i) retrieve task-specific documents, (ii) let an LLM generate or refine hypotheses, and (iii) iterate with LLM feedback. For instance, after a user supplies a seed paper and asks the LLM to generate a worthwhile hypothesis to pursue in research, Hu et al. (2024) query the Scholar API for related work, then repeatedly generate and critique hypotheses, gradually expanding a web of novel ideas. Yang et al. (2025) apply the same loop to 51 top-tier chemistry papers from 2024: experts first segment each paper into background, inspiration, and hypothesis; an LLM-based multi-agent system (MOOSE-Chem) then retrieves relevant snippets, drafts hypotheses, and scores them for originality. A similar pipeline appears in Yang et al. (2024b), where 50 conference papers are annotated in the same three fields, augmented with thematically similar web documents and 14 survey papers so that the LLM can judge both relevance and novelty.

Two variants enrich the retrieval step with structured or fine-tuned knowledge. Xiong et al. (2024) ground each hypothesis in a domain knowledge graph: entities mentioned during generation are checked against graph relations, ensuring the final claims remain fact-consistent. In contrast, Chai et al. (2024) fine-tune a T5 model (Raffel et al., 2020) on curated scientific abstracts and, during inference, retrieve citation contexts and related data; a novelty-guided loop then re-generates until the candidate is both coherent and inventive, outperforming standard transformer baselines.

**Human-in-the-loop Hypothesis Generation with LLM**: Recent studies show that combining humans with LLM support yields higher-quality, more novel hypotheses than either party working alone. Zhao et al. (2024) report that a human+LLM pipeline surpasses both human-only and LLM-only baselines. Human annotators first draft hypotheses for uncommon observations; carefully designed prompts then guide the LLM to refine each draft by adding details and improving logical flow. Low-quality hypotheses generated

by LLM are filtered with human evaluations and automated metrics such as BERTScore, and the resulting human+LLM collaboration produces the strongest hypotheses. Similarly, Si et al. (2024) involve more than 100 NLP researchers in a three-condition study: LLM-only generation, human-only generation, and LLM generation reranked by humans. Human evaluations rate the human-reranked LLM hypotheses best on Novelty, Excitement, Feasibility, and Effectiveness. Pu et al. (2025) move beyond controlled experiments by introducing IdeaSynth, a copilot-like framework that assists users throughout hypothesis formulation. When a user supplies a high-level hypothesis, IdeaSynth retrieves relevant papers through an API and summarizes key information needed for development. Users interactively edit these summaries with LLM help, adding details and improving clarity. The system then aggregates all refined nodes, employs an LLM to craft a final applicable hypothesis, and supplies suggestions for follow-up experiments.

The quality of natural-language hypothesis generation largely depends on the inherent capabilities of LLMs. Because these models excel at in-context learning, prompt strategies such as Chain-of-Thought (CoT) and Reflexion (Wei et al., 2022; Shinn et al., 2023) can be applied directly to this task. However, unlike computer-vision research, which gained rapid momentum from the ImageNet benchmark, hypothesis generation lacks a comparable, widely recognized task set. The main challenge is therefore the absence of a reliable evaluation task and benchmark for natural-language hypotheses, an issue examined further in Section 4.2.

### 4.1.2 Formal Language Hypothesis Generation with LLM

One major advantage of formal hypotheses is that once a formal language hypothesis is obtained, we can directly perform inference on it with guarantees of soundness and correctness. Depending on whether observations are represented in formal or natural language, methods for proposing a formal language hypothesis need to be discussed separately.

**Formal Language Observations**: When observations are encoded in a formal language, dedicated formal language solvers typically yield clear, white-box solutions that outperform language models. Consequently, using an LLM for these tasks is generally not preferred. Nevertheless, a few early studies in the LLM era have explored this approach. For example, Young et al. (2022) trained a transformer model on FOL abduction tasks, demonstrating that the model can generate FOL hypotheses from formal observations. Similarly, Nguyen et al. (2023) fine-tuned state-of-the-art legal transformers on FOL abduction tasks and found that models pre-trained on natural language legal abduction tasks do not show any performance improvements on FOL hypothesis generation problems.

**Natural Language Observations**: When observations are represented in natural language, traditional symbolic solvers struggle to extract the key information needed for hypothesis generation. With LLMs, however, we can directly generate formal hypotheses. A popular formal language for this purpose is code, as it offers greater flexibility than other symbolic representations like FOL, and LLMs excel at coding.

The simplest variant prompts an LLM with an observation set and asks it to produce executable functions as hypotheses that match the input-output pairs; Cheng et al. (2024) follow this pattern, treating each observation as an *(x, y)* example and evaluating the generated function by execution. Extending this idea, Wang et al. (2024); Qiu et al. (2024) have the LLM create multiple executable hypotheses, run them on the observations, feed the results back to the model, and iterate, discarding weak candidates and refining promising ones until one covers all examples. To encourage diversity, il Lee et al. (2025) first ask the model for a single-word "main concept," then use that concept to steer subsequent code generation, avoiding the similarity of low-temperature outputs and the degeneration of high-temperature sampling while still producing coherent hypotheses.

Instead of prompting an LLM to generate hypotheses from sentences, a more grounded approach is to understand how the LLM's internal representations give rise to such hypotheses. However, due to the superposition problem (Elhage et al., 2022), the same neuron may be activated for semantically unrelated concepts, making this approach challenging. Fortunately, recent work by Bricken et al. (2023) leverages sparse autoencoders (SAEs) trained on LLM activations to mitigate this issue. This enables a plausible method for directly interpreting the internal "thoughts" of LLMs via their hidden representations. Building on this line of work, Movva et al. (2025) trained an SAE on an LLM and isolated neurons activated when the model predicted the click-through rate of Twitter posts. They found that neurons associated with "surprise"

or "shock" positively contributed to the prediction, supporting the hypothesis that surprising or shocking content tends to attract more clicks.

## 4.2 Evaluation for Hypothesis Generation

Due to LLMs' strong reasoning abilities and natural language interface, many methods have been proposed for hypothesis generation, and numerous ideas based on everyday human reasoning can be adapted for this purpose (Niu et al., 2024). However, a major challenge remains in establishing a grounded and convincing way to evaluate the quality of the generated hypotheses.

### 4.2.1 Natural Language Hypothesis Evaluation

Although prompting LLMs to generate natural language hypotheses is straightforward, evaluating the quality of these hypotheses is challenging due to the ambiguity inherent in natural language representations. Consequently, a common evaluation method involves either human evaluation or using an LLM to assess the generated hypotheses' validity (Zhao et al., 2024; Yang et al., 2024b; Hu et al., 2024; Qi et al., 2024; Yang et al., 2025). While human evaluation can provide valuable insights without relying on predefined answers, it is inherently subjective, less reproducible, expensive, and sometimes not entirely convincing. Therefore, alternative evaluation strategies are needed.

**Implicit Prediction-based Evaluation**: Early benchmarks often relied on question-answering (QA) tasks that required the model to implicitly form a hypothesis to answer a question (Sinha et al., 2019; Weston et al., 2015). For example, consider the observation: *"Lily is a swan, Lily is white, Bernhard is green, Gerg is a swan. What color is Greg?"* To answer correctly, one must infer an implicit hypothesis, such as "All swans are white" or "Most swans are white," based on the fact that Lily is both a swan and white. Thus, the correct answer is "white." By verifying whether the model's answer is "white," one can indirectly assess its ability to form an appropriate hypothesis and perform reasoning. Similarly, recent work shows that prompting LLMs to generate an intermediate hypothesis and then using that hypothesis for inference yields higher performance on complex tasks (Balepur et al., 2024; Shi et al., 2023; Wang et al., 2025). However, this approach is problematic: the hypothesis may be formed incorrectly, the subsequent inference could be flawed, and the model might arrive at the correct answer through memorization or random guessing rather than proper abductive reasoning. Therefore, success in these tasks does not directly imply that the model possesses superior abductive capabilities, making them unsuitable for reliably evaluating hypothesis generation.

**Ground Truth-based Evaluation**: Some studies build benchmarks with labeled hypotheses so that outputs of LLM can be matched directly against references. DEER (Yang et al., 2024a) supplies 1,200 fact–rule pairs, all written in natural language by experts across six topics-zoology, botany, geology, astronomy, history, and physics. Generated hypotheses are compared with the gold rules using token-level mapping metrics like METEOR (Banerjee & Lavie, 2005). In biomedicine, Qi et al. (2024) curate a benchmark with both seen and unseen samples: the seen split contains 2,700 background–hypothesis pairs collected before January 2023, whereas the unseen split has 200 pairs collected after that date. Outputs are evaluated against the ground truth with BLEU and ROUGE (Papineni et al., 2002; Lin, 2004). On synthetic corpora such as WIKI and BILLS (Pham et al., 2024; Zhong et al., 2024), Movva et al. (2025) treat hypothesis generation as identifying the key features that drive a prediction. The model proposes feature sets, which are judged by how well they match and cover the ground-truth features, thereby quantifying the LLM's ability to isolate causal signals.

Despite these efforts, Yang et al. (2024b) note that reference-based metrics such as BLEU, ROUGE, and METEOR assume a single correct answer and therefore struggle to capture the open-ended nature of hypothesis generation; developing fair, reliable metrics remains an open challenge.

### 4.2.2 Formal Language Hypothesis Evaluation

Unlike natural language hypotheses, formal hypotheses evaluations are more grounded due to their clarity and unambiguous semantics.

**Prediction based Hypothesis validation**

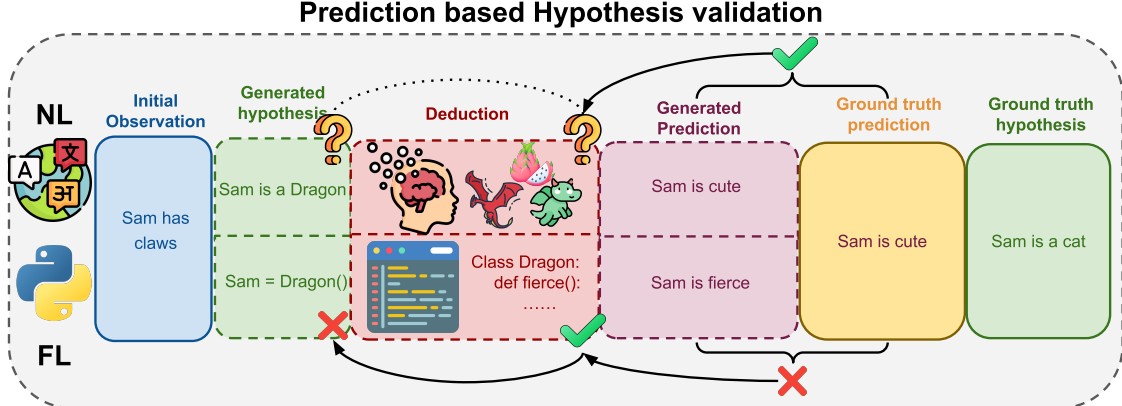

Figure 4: Directly evaluating a hypothesis is often impractical. A reliable alternative is prediction-based evaluation, which operates on a key principle of formal logic: deductive reasoning from a correct hypothesis must yield correct predictions. Therefore, a single incorrect prediction is sufficient to definitively falsify the hypothesis. However, this reliability fails for non-deterministic LLMs evaluating NL hypotheses. The process is fundamentally flawed because an incorrect hypothesis can still generate a correct prediction.

**Ground Truth-based Evaluation**: Generated formal hypotheses can be evaluated against pre-defined ground truth hypotheses. Unlike natural language evaluation, where ground truth is often written by domain experts and evaluated using token-level metrics like BLEU or ROUGE, formal hypotheses can be evaluated procedurally using solvers. This allows us to verify correctness deterministically. For example, Bowen et al. (2024) designed formal representations for synthetic grouping tasks to evaluate formal language hypothesis generation. Hua et al. (2025) constructed their benchmark based on deterministic regular functions, providing a procedural framework for evaluating formal hypotheses. Similarly, Young et al. (2022) used FOL representations, where LLMs were tasked with generating FOL hypotheses to explain given facts, and the outputs were evaluated by comparing them against ground truth hypotheses verified by solvers.

**Prediction-based Evaluation**: Since inference on formal hypotheses is deterministic, a common evaluation method is to test whether the generated hypothesis produces correct outcomes on held-out examples (See in Figure 4). For instance, Rule (2020) propose the *list function task*, where LLMs generate a hypothesis function from observed (x, y) pairs, and evaluation is based on how well the hypothesis predicts hidden pairs. Similarly, Chollet (2019) introduces the Abstraction and Reasoning Corpus (ARC), where tasks involve transforming input grids of colored cells into output grids. The generated function is executed on test inputs, and correctness is determined by exact matches with the target output grids, including grid dimensions. Liu et al. (2024) further propose a benchmark consisting of arithmetic calculations, color token mapping, and Kalamang vocabulary tasks, all evaluated in the same way. Additionally, Li et al. (2025a) construct diverse application scenarios, including list transformations, real-world problems, code generation, and string transformations, where the generated hypothesis is executed on both seen and test observations and the final score aggregates performance across both sets. In a more realistic setting, Chen et al. (2024) extract 102 tasks from 44 peer-reviewed publications. Based on a concise instruction and the corresponding dataset for each task, the agent must generate a formal hypothesis—in the form of a self-contained Python program—that proposes a complete methodology to reproduce the original scientific analysis. The validity of this generated hypothesis is then directly evaluated against expert-validated test cases and through fine-grained, rubric-based human evaluations.

While the output of formal hypothesis generation can be an executable program, making it procedurally similar to program synthesis, the two paradigms are distinguished by their fundamental objectives. Program synthesis addresses computationally concrete problems where the specifications are clearly defined and the causal relationship between inputs and outputs is explicit. The core challenge is one of implementation—writing code to satisfy a known requirement. In contrast, formal hypothesis generation confronts holistic, semantically rich scenarios where the underlying rules are unknown. Here, the central challenge is interpretation and discovery, a crucial step that precedes any implementation. The agent must first

analyze a phenomenon, abstract the necessary features from information-rich inputs using domain knowledge and commonsense, and formulate the problem itself. For instance, even with structured inputs like the ARC benchmark, success hinges on first deducing the semantic logic—which patterns are relevant and how—before a transformation can be coded. Therefore, formal hypothesis generation crucially includes the preliminary scientific task of discovering an unknown rule, a step that is already assumed to be complete in the well-defined world of program synthesis.

### 4.3 Discussion and Future Directions in Hypothesis Generation

There exists a significant gap between formal and natural language approaches to hypothesis generation. In natural language hypothesis generation, observations typically originate from recent research papers, and generated hypotheses can potentially inspire novel research ideas with tangible real-world impacts (Eger et al., 2025). However, rigorous and reliable evaluation methods for such hypotheses remain underdeveloped. Token-based metrics, such as BLEU or ROUGE, do not effectively capture the qualitative aspects of open-ended hypothesis generation (Yang et al., 2024b). Meanwhile, alternative approaches involving human or LLM-based evaluations are costly, subjective, and prone to inconsistencies.

Conversely, formal language hypothesis generation benefits from grounded, objective evaluation methods. Nevertheless, existing formal tasks often involve simplified or artificial scenarios that fail to reflect the complexity and nuance inherent in real-world applications. Consequently, the field faces a trade-off: formal representations facilitate robust evaluation but risk omitting critical real-world nuances, while natural language representations capture real-world complexity yet lack rigorous evaluation mechanisms.

To address this challenge, future research in hypothesis generation could focus on two key directions. Firstly, there is an urgent need to develop novel evaluation methodologies for natural language hypothesis generation. Current methods, such as implicit prediction and ground-truth comparisons using token-level similarity, are inadequate. Alternative strategies like multi-dimensional human assessments, structured feedback, and hybrid evaluation frameworks should be explored. A deeper insight gained from our survey is that recent evaluation methods are built upon an easy-to-overlook assumption: that for any given problem, there exists a single "golden" hypothesis. Consequently, evaluation is performed by directly comparing a model's output to this predefined "golden" hypothesis or to the conclusions drawn from it. This approach is not aligned with scientific practice. In the real world, many published hypotheses based on limited observations are ultimately proven wrong, and many deviate from what is known to be "correct" today. We cannot, however, simply classify the abductive process that generates these hypotheses as flawed or incorrect. Furthermore, different perspectives can give rise to multiple, semantically different, yet plausible hypotheses. It is a mistake to conclude that a model lacks strong abductive reasoning simply because it generates a plausible hypothesis that doesn't match the predefined "golden" answer. Therefore, before designing new benchmarks, it is essential to first step back and establish a clearer definition of what constitutes a "good hypothesis" and "good abductive reasoning," based on how hypothesis is generated by experts in practice. Secondly, bridging the gap between formal and natural language hypothesis generation is crucial. Leveraging code as an intermediate representation is a promising path forward, as it combines evaluative rigor with expressive capability. However, existing code-based benchmarks often focus on oversimplified problems that lack relevance to practical scenarios. Thus, a compelling direction for future research is to develop realistic, code-based hypothesis-generation tasks grounded in established research papers, real-world datasets, and open-source repositories.

## 5 Hypothesis Application with Deduction

Given a hypothesis $h$, hypothesis application is defined as the derivation of a new observation $o_{\text{new}}$ such that:

$$h \models o_{\text{new}}$$

In some cases, the hypothesis may depend on a context $c$, so that $h$ can be viewed as a function of $c$. In this context-dependent formulation, hypothesis application is defined as deriving a new observation $o_{\text{new}}$ such that:

$$h(c) = o_{\text{new}}$$

In our work, we follow Peirce's definitions for reasoning. Accordingly, the primary process used in hypothesis application is deduction, the method of deriving necessary consequences from a given hypothesis.

Notably, while much research trains LLMs on formal language corpora to improve their general reasoning abilities, this survey focuses on how LLMs can advance the hypothesis discovery. For the specific task of applying a formal hypothesis, a dedicated solver (e.g., a FOL prover) is the superior tool, as its deterministic nature guarantees a correct and sound prediction. A probabilistic LLM can, at best, only be trained to approximate the performance of a perfect solver in this closed setting; it cannot be more correct or offer a better solution. Therefore, since a reliable and transparent method already exists, there is little motivation to use an LLM for deductive application of formal hypotheses. This section, consequently, will focus on the application and evaluation of hypotheses in natural language.

## 5.1 Method

**LLM as autoformalizer:** Since formal symbolic solvers yield sound and correct predictions, Pan et al. (2023); Olausson et al. (2023); Kalyanpur et al. (2024) treat LLMs as autoformalizers, using them to translate natural language hypotheses into formal representations like FOL and code before applying a formal inference procedure. This translation significantly improves deductive correctness. However, these methods have primarily been evaluated on benchmarks such as ProofWriter (Tafjord et al., 2021) and FOLIO (Han et al., 2024), where the questions are already closely aligned with formal language. For example, given the input *"Fact1: Eric is young, Fact2: Dave is white, Rule 10: if someone is young and not kind then they are big"*, translating this into FOL is relatively straightforward. It remains unclear whether LLMs can reliably parse more complex, everyday natural language into formal representations.

**Fine-Tuning-Based Method:** Fine-tuning is a common approach to improve model performance when corresponding training data is available. Sun et al. (2024) proposed a synthetic "StringGame" task in which ground truth hypotheses and answers are provided. Leveraging the CoT approach, a LLM is prompted to generate multiple candidate hypothesis application trajectories along with their results. By comparing these results with the ground truth, the trajectories that produce correct outcomes are identified as correct and stored for fine-tuning. The resulting fine-tuned model then demonstrates improved performance in both hypothesis application and instruction following.

**Prompt-Based Method:** Although CoT prompting has improved performance on multi-hop question answering tasks, Sun et al. (2024) found that it does not directly enhance performance in hypothesis application. Therefore, new prompting methods have been designed specifically for this purpose. Inspired by mathematical induction, Cai et al. (2025) propose quantifying the difficulty of a question so that the LLM can solve it incrementally, from simpler versions to more complex ones, ultimately arriving at the correct answer. In another approach, Ling et al. (2023) design a pipeline that supervise the correctness of each reasoning step during hypothesis application. First, the LLM indexes all premises; then it is asked to label the minimal set of premises required to derive new facts. This pipeline generates multiple candidate hypothesis application trajectories, and by having the LLM vote on each step, the most convincing deductive trajectory is selected.

## 5.2 Evaluation for Hypothesis Application

Although many benchmarks and evaluation methods exist for general deductive reasoning, such as question answering and mathematical tasks like GSM-8k (Cobbe et al., 2021), these question types do not explicitly test the formation of new facts based on given hypotheses or rules. Evaluating the correctness of a natural-language deductive trajectory is challenging because annotated reasoning paths for hypothesis application are scarce, and the same result can follow from different reasoning paths. As a result, most evaluations use prediction-based checks. We assume that, given a correct hypothesis and a known ground-truth result, a valid deduction will reproduce that result. By comparing the model's deduced outcome with the ground truth, we can judge whether its deduction is correct. For example, take the hypothesis *"Coin flips are independent and identically distributed (i.i.d.) with a 50 percent chance of heads."* When asked, *"After three consecutive heads, what is the probability of a tail on the fourth flip?,"* a flawed model might claim the chance of a tail has increased. In fact, under the i.i.d. assumption, the probability remains 50 percent. Supplying

the correct hypothesis and comparing the model's answer to the true result lets us evaluate whether its deductive reasoning is valid.

Several benchmarks have been developed to evaluate hypothesis application. Inspired by the "Turtle Soup" game, Yu et al. (2024b) created TURTLEBENCH, where an LLM must answer "True," "False," or "Not Relevant" to a series of questions to test its ability to follow a story. The RULES benchmark, introduced by Mu et al. (2024), provides 14 rule-following scenarios with programmatic evaluation to objectively assess adherence to specified rules. Expanding on complexity, the Holiday Puzzle benchmark (Cai et al., 2025) features multiple holiday schedule scenarios and uses test cases to verify correct computation. To specifically test inferential rule-following, Sun et al. (2024) constructed RuleBench, which evaluates a model's ability to apply both factual and counterfactual rules; their experiments reveal that while LLMs achieve near-perfect accuracy on factual rules, their performance drops dramatically under counterfactual scenarios, revealing a significant capability gap.

### 5.3 Discussion and Future Directions in Hypothesis Application

While traditional deductive reasoning tasks (e.g., question answering, problem solving) in LLMs have been widely studied, the capability for hypothesis application remains significantly underexplored. According to Sun et al. (2024), hypothesis application involves inferential rule-following, requiring models to consistently apply given hypotheses to derive novel knowledge in unfamiliar domains. Robust hypothesis application is critical to hypothesis discovery, as hypotheses must generalize to scenarios with unseen observations. However, existing LLMs frequently struggle to extend hypotheses beyond familiar contexts, thus limiting the evaluation of hypothesis generation.

Future research could therefore focus on rigorously evaluating LLMs' hypothesis application, both factual and counterfactual, in novel scenarios. Developing benchmarks explicitly designed for hypothesis-driven inference in unfamiliar domains could reveal important insights into model adaptability and generalization. Additionally, current evaluations of hypothesis application mainly rely on outcome-based correctness, comparing predicted results to ground truth given correct hypotheses. However, incorrect reasoning may still lead to correct predictions in natural-language contexts. Although Ling et al. (2023) propose improving hypothesis application by intervening in reasoning trajectories, a large-scale benchmark specifically designed to evaluate trajectory-based hypothesis application remains absent.

Although as stated in the beginning of the section, if all premises and hypothesis need to be applied are already represented in formal language like FOL or code. The hypothesis application tasks with formal language do not bring gain with LLM. There is still cases where LLM may help improve this problem

## 6 Hypothesis Validation with Induction

According to Peirce, induction validates a hypothesis by updating its confidence when new evidence appears. However, in studies that focus exclusively on induction, tasks are typically one-off: a hypothesis (or set of hypotheses) and a collection of observations are provided, and there is no iterative updating of confidence. A simplified framework for hypothesis validation treats it as a multiple-choice problem: given observations $\mathbb{O} = \{o_1, o_2, \ldots, o_n\}$ and a set of hypothesis $\mathbb{H} = \{h_1, h_2, \ldots, h_m\}$, the model selects the most possible hypothesis. In simpler scenarios, where only one hypothesis is provided, the model determines whether the hypothesis correctly explains the observations. In the next section, when combined with deduction and abduction, induction can subsequently be used to iteratively update the confidence in the hypothesis.

Natural language representations add significant complexity to induction. In formal language settings, all necessary information is explicitly provided, and reasoning follows rigorous, well-defined steps. In contrast, validating a natural language hypothesis often requires commonsense knowledge and interpretation of nuanced language. For example, consider the observations "Neil wanted to see the mountains of Asia" and "Neil loved being so close to the mountains in Nepal," with candidate hypotheses "Neil booked a trip online" and "Neil took a trip to see the Rocky Mountains instead." Here, the nuanced meaning of the term "instead" and the geographic relationships require careful analysis and may lead to different conclusions. Indeed, Zhang et al. (2020) reports that, when verifying their dataset where five annotators judged the plausibility of hand-

written hypotheses, disagreements occurred in 62.34% of 1,365 explanations, underscoring the challenge of natural language hypothesis validation.

## 6.1 Method

### 6.1.1 Formal Language Hypothesis Validation

He & Lu (2024) introduce the CauseJudger framework, which leverages LLMs at every stage to validate candidate hypotheses. First, an LLM transforms the natural language inputs into an FOL-based representation by integrating each candidate hypothesis into the premises. Next, an LLM filters out irrelevant premises and rules. Finally, another LLM performs forward reasoning to decide which hypothesis explains the observations.

### 6.1.2 Natural Language Hypothesis Validation

**Prompt-Based Method:** Lampinen et al. (2022); Sun et al. (2024) employ a few-shot prompting approach for hypothesis validation. In this method, case triplets, consisting of an observation, a hypothesis, and its corresponding validity, are provided to the model, which then answers a hypothesis validation question. Although this approach improves performance, Sun et al. (2024) reports that the performance boost is limited. Their experiments further indicate that fine-tuning outperforms few-shot prompting.

**Fine-Tuning-Based Method:** Since hypothesis validation essentially constitutes a classification problem, many Natural Language Inference (NLI) datasets can be adapted into hypothesis validation tasks. Consequently, fine-tuning is a popular method in this context. For example, Zhao et al. (2023); Chan et al. (2023); Sun et al. (2024) fine-tune models to select the correct hypothesis from a set of hypotheses based on new observations.

## 6.2 Evaluation for Hypothesis Validation

### 6.2.1 Formal Language Evaluation

Along with the CauseJudger framework, He & Lu (2024) also proposed the CauseLogics dataset. Based on the required formal reasoning depth, the dataset is divided into four difficulty levels for hypothesis validation tasks, with 50,000 samples per level. Each hypothesis is assigned a binary ground-truth label indicating whether it correctly explains the observations.

### 6.2.2 Natural language Evaluation

**Binary-Classification-Based Evaluation:** Lampinen et al. (2022) chose a subset of 40 tasks from the crowd-sourced benchmark BIG-bench (bench authors, 2023) and constructed their own benchmark specifically for hypothesis validation. Each data sample consists of an observation, its corresponding hypothesis, and a ground truth label indicating whether the hypothesis truly explains the observation.

Hypothesis validation using natural language is inherently challenging because the implicit information and required common-sense background are not explicitly stated. This often leads different individuals to draw different conclusions when validating a hypothesis based solely on recalled information. Rudinger et al. (2020) mitigate this issue by adopting a different strategy. Instead of asking annotators to directly judge whether an observation explains a hypothesis, they ask the model to determine if a given observation weakens or strengthens the hypothesis. Specifically, they sample observation–hypothesis pairs from existing datasets and then manually craft two types of sentences: one that acts as a "strengthener" (increasing the likelihood of the hypothesis) and one that acts as a "weakener" (decreasing the likelihood of the hypothesis). A quality check of the labels confirmed high inter-annotator agreement on the strengthening and weakening effects. During evaluation, the model is required to decide whether a new observation strengthens or weakens the hypothesis. This approach aligns with the paper's goal of modeling defeasible inference by leveraging explicit contextual updates rather than relying on potentially variable human interpretations of implicit information. Furthermore, Zhang et al. (2025) extended this task to include visual observations. In their extension, given

a visual observation and a natural language hypothesis, an LLM is tasked to determine whether the provided sentence serves as a strengthener or a weakener.

**Multiple-Choice-Based Evaluation:** Bhagavatula et al. (2020) introduce the ART benchmark, comprising roughly 20k narrative contexts where each sample includes two time-ordered observations, one depicting a story's start ($o_1$) and the other its outcome ($o_2$), alongside two hypotheses: a plausible explanation ($h^+$) and a less plausible one ($h^-$), challenging models to choose the best explanatory hypothesis and enabling adaptation to hypothesis-generation tasks evaluated against ground-truth explanations. Similarly, Jiang et al. (2023) present the BRAINTEASER benchmark of about 1.1k lateral-thinking puzzles, each offering a question with multiple-choice answers, one that defies commonsense and several conventional distractors, in both sentence (narrative) and word (meaning-alteration) formats to test creative reasoning, with additional semantic and context reconstruction variants assessing reasoning consistency and robustness across formulations. Moreover, Del & Fishel (2023) introduced the True Detective benchmark for deep hypothesis validation, featuring 191 long-form detective puzzles ($\approx 1200$ words each) from the "5 Minute Mystery" platform, where models (and humans) select the correct explanation from 4–5 options, human accuracy averages 47%, top solvers exceed 80%, and each puzzle includes golden chain-of-thought explanations detailing the reasoning steps that lead to the correct answer.

### 6.3 Discussion and Future Directions in Hypothesis Validation

Previous literature often conflates hypothesis generation and hypothesis validation, primarily due to ambiguity inherent in the inference to the best explanation (IBE) paradigm. Within IBE-based approaches, hypothesis validation typically appears as an implicit intermediate step, where selecting the "best" hypothesis is frequently based on unclear or subjective criteria without dedicated, independent evaluation. However, adopting Peirce's explicit distinction between abduction, deduction, and induction clearly separates validation from generation, underscoring the need for dedicated research on validating hypotheses against newly observed evidence.

Current validation methodologies predominantly adopt end-to-end metrics that only assess final correctness, neglecting the reasoning processes and commonsense knowledge required to validate hypotheses in realistic settings. The subjective nature of natural language, coupled with different interpretations of observations, highlights the necessity for richer evaluative frameworks. Future benchmarks should incorporate detailed intermediate Chain-of-Thought data, capturing explicit reasoning steps humans take when validating hypotheses, such as recalling relevant commonsense knowledge and performing nuanced inference. Evaluations should then emphasize consistency between the reasoning process and available commonsense context rather than relying solely on superficial similarity to reference answers. Such benchmarks would greatly enhance our understanding of hypothesis validation and better reflect the complexities of human-like reasoning.

## 7 Hypothesis Discovery

Although many works introduced in the previous sections propose methods and evaluation metrics, they mainly focus on individual phases of **Hypothesis Discovery**—Hypothesis generation *(Abduction 4)*, Hypothesis application *(Deduction 5)*, and Hypothesis validation *(Induction 6)*. However, in real-life **Hypothesis Discovery**, these reasoning stages are not independent and must be treated holistically. Initially, we form hypotheses based on limited observations using abduction, which subsequently informs the application of these hypotheses through deduction, enabling the collection of further evidence. Concurrently, induction continuously evaluates and resolves inconsistencies arising between newly obtained observations and earlier hypotheses. This iterative interplay means that each hypothesis formulated, action taken, observation gathered, and inconsistency identified dynamically shapes and reshapes our evolving understanding, influencing subsequent reasoning steps and contributing to diverse interpretations of the world. Treating any single reasoning phase in isolation oversimplifies hypothesis discovery. For example, although Bowen et al. (2024) evaluated every reasoning, they handled each step separately and thus failed to assess the true rule-learning capability of LLMs. Consequently, integrating abduction, deduction, and induction into a unified learning loop remains both challenging and largely understudied, yet it is the ultimate goal for constructing end-to-end agents capable of scientific discovery.

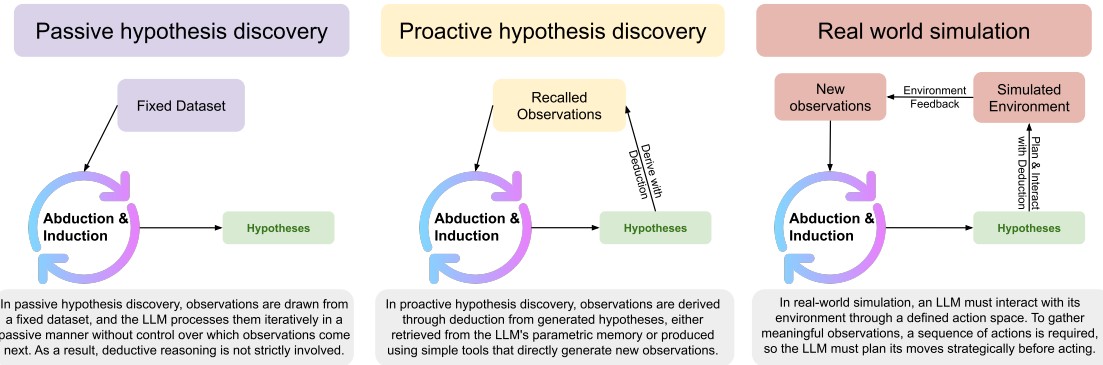

Figure 5: Differences and similarities among different types of hypothesis discovery tasks

Despite a few studies that acknowledge the interdependence among reasoning types and allow models to refine hypotheses iteratively, they still overlook two decisive aspects of real-world hypothesis discovery. First, most benchmarks remain static and passive: they hand agents a fixed set of observations deemed sufficient to reach the correct hypothesis, whereas real-life hypothesis discovery requires actively seeking additional evidence. Second, even in settings that allow proactive information gathering, the granularity of the action space is still too coarse: agents fetch observations via one-shot "recall" or "web-search" commands, whereas real scientists must strategically plan and carry out precisely staged experiments—often designing specialized equipment at each step. Recognizing these limitations, we categorize existing hypothesis-discovery research into three classes (see Fig. 5).

## 7.1 Passive Hypothesis Discovery

In this type of study, LLMs generate, apply, and validate hypotheses iteratively. However, the observations are provided by a fixed dataset. The LLM does not need to worry about which observations it will receive. Instead, it simply reasons based on the given data, passively receiving and processing the information provided.

Zhu et al. (2024) proposed the Hypotheses-to-Theories (HtT) Framework to generate formal hypotheses (e.g., *"if A then B"*) by leveraging existing benchmarks (Sinha et al., 2019; Wang et al., 2022b; Rule, 2020). In HtT, LLMs generate a hypothesis and propose learned rules to solve each question. When a new question is received, the model first formulates a preliminary hypothesis based on the context. It then proposes candidate rules that might lead to the correct answer. These candidate rules are applied to the problem and verified against the ground truth. Rules that consistently yield correct predictions are retained and added to the rule library, while ineffective ones are discarded. Iteratively, after processing all questions in the benchmark, the LLM builds a rule library containing effective rules for solving the questions.

Zhou et al. (2024) proposed the HypoGeniC framework. Unlike HtT, HypoGeniC is evaluated on more realistic datasets such as Shoe Sales, Deceptive Reviews (Granhag & Vrij, 2005), Headline Popularity (Matias et al., 2021), and Tweet Popularity (Tan et al., 2014). Due to the complexity of real-world data, the generated hypotheses are more nuanced and expressed in natural language. Similar to HtT, HypoGeniC begins by generating a set of candidate hypotheses from a small number of examples. As new observations are processed, each hypothesis is used to make predictions and is assigned a reward based on its accuracy. The system dynamically updates the confidence of each hypothesis; those that consistently perform poorly are removed from the hypothesis bank. New hypotheses are generated from examples that existing hypotheses fail to explain, allowing the model to refine and expand its understanding over time.

Both HypoGeniC and HtT simplify hypothesis discovery by relying on benchmark questions that include ground-truth answers. This configuration allows an external algorithm, not the LLMs themselves, to validate generated hypotheses and update their confidence based on the correctness of predictions. In real-world

scenarios, where no ground-truth answers are available, these frameworks become inapplicable and would require substantial adaptation.

## 7.2 Proactive Hypothesis Discovery

In real-life hypothesis discovery, we do not start with a predefined set of observations that continuously propose new insights. Instead, once an initial hypothesis is formed, we proactively recall our memories or explore further to gather new observations that either strengthen or weaken the hypothesis, allowing us to verify and refine our ideas.

Given a hypothesis, Li et al. (2024) and Jung et al. (2022) propose two proactive methods for hypothesis discovery that both leverage the LLM's parametric memory to generate evidence that either strengthens or weakens the hypothesis. In Hypothesis Testing Prompting, the model directly uses its internal reasoning to evaluate the generated evidence, determining which pieces are more convincing, and then decides whether the hypothesis is correct based on the balance of evidence that strengthens or weakens it. In contrast, Maieutic Prompting iteratively constructs a tree of evidence by generating both strengthening and weakening explanations. It then employs the LLM to assign a belief score (reflecting the model's confidence in the evidence) and a consistency score (measuring how well the evidence aligns with the hypothesis). Finally, a MAX-SAT solver is applied to select the subset of evidence that maximizes the overall scores, thereby determining whether to accept or reject the hypothesis.

Different from relying solely on an LLM's parametric memory to generate new evidence, Seals & Shalin (2024) propose a minimal setting for proactive hypothesis discovery. Inspired by the Wason Task from cognitive science, this task challenges LLMs to prove a formal language hypothesis of the form "if $p$ then $q$." Here, both $p$ and $q$ are objects described in natural language, for example, "if a person is a man, then he drinks alcohol." The task provides four cards, each with two sides representing different attributes. Initially, one side of each card is shown, displaying $p, q, \neg p$, and $\neg q$, while the other side reveals the state of another attribute. To rigorously validate the hypothesis "if $p$ then $q$," one must flip the $p$ card to confirm that its hidden side is $q$ (modus ponens) and flip the $\neg q$ card to check that its hidden side is $\neg p$ (modus tollens). Flipping only these two cards provides sufficient evidence for the hypothesis, while the other two cards do not offer the necessary information. Thus, in this benchmark, by proactively flipping two cards, we can determine whether the LLM can correctly identify natural language expressions of $p$ and $q$ and validate the hypothesis using a minimal action space.

Moreover, Conti et al. (2024) propose APEx, a multimodal automatic benchmarking framework that evaluates hypotheses about large multimodal models in a fully automated and iterative fashion. For example, to test a hypothesis such as "a model is able to identify graffiti-styled images," APEx first leverages text-to-image retrieval and generation tools to create a tailored set of test images. It then employs a range of transformation tools to perform image augmentation, introducing variations that challenge the models' robustness. In an iterative experimental loop, the framework executes these experiments on a library of models, analyzes the results, and refines the testing protocol accordingly.

## 7.3 Complete Loop: Real-World Hypothesis Discovery Simulation

Other works equip LLM agents with interactive environments that more closely mirror the complexity of real-world hypothesis discovery by combining planning, acting, and evidence collection. For example, Xu et al. (2023) construct a Minecraft–like world in which a "vandal" agent performs up to 26 types of actions (e.g., moving, eating, crafting) to achieve a hidden goal (such as collecting lava or crafting a particular item) and leaves behind tracks as evidence. A detective agent—driven by reinforcement learning to maximize information gain—then gathers those tracks and presents them to an LLM, which must answer a multiple-choice question about the vandal's original objective. Because evidence collection relies on an RL policy rather than LLM planning, this setup evaluates only the model's capacity to interpret evidence, not its ability to proactively generate and test hypotheses in a dynamic setting.

Building on this approach, Wang et al. (2022a) introduce 30 scientific tasks drawn from five topics in fifth-grade curricula, ranging from measuring the friction coefficient of an inclined plane to testing electrical

conductivity. Here, agents must execute long action sequences and apply deductive reasoning grounded in established theories and definitions to complete each task. Likewise, Jansen et al. (2024) design 120 experiments across eight subjects (e.g., Chemistry, Archaeology), each with three difficulty levels, and allow 14 coarse-grained actions (such as "take," "put," and "move"). Agents are evaluated on (1) task completion, (2) execution of key experimental steps, and (3) accurate hypothesis discovery compared to a ground truth. While these virtual labs simulate multi-step procedures and test hypothesis application, their restricted action spaces support only qualitative inference and preclude the fine-grained interventions needed for quantitative rule-learning.

To address these limitations, He et al. (2024) propose puzzle environments in which agents can input arbitrary integers or letters and receive tailored feedback based on a hidden rule. In this framework, an LLM must iteratively probe the environment, uncover the underlying quantitative rule, and solve the puzzle. Performance is assessed not only by whether the agent solves the puzzle but also by human judgments of the clarity and rigor of its reasoning steps, thereby offering a finer-grained evaluation of both quantitative hypothesis generation and the quality of the model's deductive process.

### 7.4 Discussion and Future Directions in Hypothesis Discovery

Hypothesis discovery fundamentally differs from isolated reasoning tasks by requiring iterative learning and continuous refinement of hypotheses within dynamic, evolving contexts. Particularly in Real-World simulation scenarios, the decisions and actions taken by an LLM may lead to entirely different trajectories of observation collection, varied learning efficiencies, and alternative hypotheses.

Building effective benchmarks for hypothesis discovery requires constructing rich, realistic environments capable of simulating real-world complexities. These environments should contain diverse, comprehensive action spaces and varied observational feedback mechanisms. Compared to traditional static, label-based datasets, creating such benchmarks is significantly more labor-intensive, demanding at least two key components: 1, **A set of rules unknown to the LLM** that can be learned within the environment. 2, **A sufficiently expressive action space** that allows the LLM to interact with the environment, receive feedback, and gather new information. Based on our survey, we provide the following insights to satisfy these two components.

Firstly, since current LLMs are already trained on a vast corpus of text, finding a rule that is unknown to them is tricky. Based on our survey, we have identified two valid methods. One approach, as proposed by He et al. (2024); Jansen et al. (2024), is to use self-defined, fictitious rules. These rules are novel to LLMs; however, this introduces limitations. The handmade, fictitious rules are less complex compared to real-world rules. A second method is to test LLMs on newly founded hypotheses from the most recent work, which would not have been part of the LLM's training corpus (Qi et al., 2024). However, although this approach eliminates the rule leakage problem, it introduces unique challenges. As previously noted, the work by Qi et al. (2024) is limited to hypothesis generation; additional work is needed to adapt suitable research into interactive environments for hypothesis discovery.

Secondly, regarding a sufficiently expressive action space, it is costly to construct an interactive environment from scratch. Luckily, a recent hot topic involves leveraging LLM code agents as researchers to reproduce the code and methods in new research papers (Huang et al., 2023; Chan et al., 2024; Yan et al., 2025a). Although the aim of these studies is limited to reproducing the code program (similar to program synthesis), this presents a possible and valuable direction. One could translate an annotated paper into an interactive environment. It would not be necessary to define the interactions, as the code generated by the LLM agent already provides a sufficiently fine-grained and expressive action space. One would then mask the hypothesis and results of the paper and test whether an LLM can reproduce the statements and hypotheses proposed.

## 8 Summary

In this survey, we have presented a comprehensive and structured framework for hypothesis discovery using LLMs, guided by Peirce's reasoning paradigm of abduction, deduction, and induction. We systematically ex-

plored current methods and benchmarks across the three core components: hypothesis generation, hypothesis application, and hypothesis validation.

A central finding of our analysis is the significant trade-off between formal (FL) and natural language (NL) representations. While FL enables rigorous, objective evaluation, it is often confined to simplified, artificial scenarios that lack real-world complexity. Conversely, NL captures the nuanced complexity of real-world reasoning but suffers from a critical lack of reliable and reproducible evaluation metrics. Most fundamentally, current evaluation paradigms are often biased by the underlying assumption of a single "golden" hypothesis, which ignores that multiple, semantically distinct theories can be plausible and valid—a stark deviation from authentic scientific practice.

Furthermore, we advocate for a paradigm shift away from isolated tasks and toward integrated, dynamic environments that faithfully model the scientific process. Building such benchmarks requires two key features: a set of rules genuinely unknown to the LLM and a sufficiently expressive action space for interaction. A promising path forward involves creating these environments by adapting recent research papers, whose findings post-date the LLM's training corpus. This strategy ensures the underlying scientific principles are genuinely unknown to the model. Within this setup, the LLM agent's objective is to rediscover the paper's central hypothesis by generating code to conduct experiments. The act of code generation itself serves as a naturally expressive and fine-grained action space, shifting the focus from mere code reproduction to genuine hypothesis discovery.

By developing these holistic environments that demand proactive hypothesis generation, robust application, and continuous validation against evolving evidence, future research will significantly advance the ability of LLMs to not merely execute instructions but to autonomously generate, refine, and validate hypotheses, thus realizing their potential as true engines of discovery and innovation.

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
