# OpenReview forum: "From Reasoning to Learning: A Survey on Hypothesis Discovery and Rule Learning with Large Language Models"
_TMLR — Accepted by TMLR_

### Review · Reviewer_ywQH · 2025-06-16

**Summary Of Contributions:**

The paper forms an exhaustive survey of what the authors collectively label as "hypothesis discovery" and "rule learning" in systems that are predominantly driven by a language model. The authors do a good job of synthesizing results from across the LLM reasoning literature, managing to achieve quite a bit of breadth without ever making me feel like they had to "shoehorn" a paper to fit into their framework. The primary contributions of the paper are thus:
- Collecting and summarizing a large number of results from the very fast-moving (and, let's be honest, rather disorganized) LLM reasoning literature
- Identifying common strengths and weaknesses across different papers from the literature in terms of how far they go to evaluate LLM's ability to discover hypotheses and learn rules
- Presenting a unified framework through which to view progress in the field. Though the framework (by Pierce) itself is well-established in the philosophy of logic, I think it is a novel and interesting way to frame research in this field.

**Audience:**

Yes

**Broader Impact Concerns:**

None.

**Claims And Evidence:**

Yes

**Requested Changes:**

- (Not critical): Consider whether adding more visual summarizations of the survey would help the user digest your insights. These could potentially go in the appendix if you do not want to lengthen the paper itself too much.

**Strengths And Weaknesses:**

Strengths:
- The breadth of the survey is appropriate. The authors have managed to include all the major relevant works in the field which I am aware of, as well as some more obscure ones that nicely complement their results. At the same time, I never got the sense that they were including "throw-away" citations just to pump the numbers up.
- The paper is exceptionally well written from a linguistic point of view.
- The structure of the paper makes it easy to follow. Indeed I think it is testament to the strength of the paper that within the first 5 minutes of skimming the manuscript I felt like I had already developed a more structured mental model of the literature.
- The use of Pierce's framework of hypothesis discovery shines a new light on recent application-focused research in the LLM literature. I believe practitioners and researchers alike will find themselves coming up with plenty of new ideas just by reading this survey.

Weaknesses:
- My only complaint is that the survey is quite light on figures and tables, which might make the paper a bit more digestible. Luckily the text is written so well that it nonetheless remains quite manageable to simply read it in full.

---

> ### Author Response · Authors · 2025-07-11
> **Feedback to reviewer ywQH**
>
> We thank Reviewer 3 for their supportive and constructive review. We are pleased that you found the paper’s structure and writing clear, and that Peirce’s framework offered a helpful lens on the literature.
>
> You suggested adding more visual summaries, and we agree this makes our insights easier to grasp. We have therefore made the following revisions:
> - Figure 1: The leaves are now color-coded by section to enhance visual clarity.
> - Figure 3 (Section 2.1): Newly added to clarify the distinctions among hypotheses, theories, and rules.
> - Figure 4 (Section 4.2.2): Newly added to illustrate why prediction-based evaluation is problematic for natural-language hypotheses yet appropriate for formal-language hypotheses.

---

> > ### Comment · Reviewer_ywQH · 2025-07-22
> > **Response to authors**
> >
> > I thank the authors for engaging with my comments (and those of the other reviewers) constructively. I have reviewed the changes made in the updated PDF and in addition to the welcome addition of several new figures, the changes made to some aspects of the discussion (as requested by G3US, in particular) has clarified some nuanced of the discussion further. Although I do not agree with 1sKv that the summary needed to be overhauled, the new version seems fine to me. Overall, I am very happy with the state of the paper.

---

### Review · Reviewer_1sKv · 2025-06-30

**Summary Of Contributions:**

This work takes a comprehensive survey of knowledge discovery. The paper structure is clear. And this work review is enough paper for each section. In the future, the paper could help summarize the progress of hypothesis discovery and rule learning with large language models.

**Audience:**

Yes

**Claims And Evidence:**

Yes

**Requested Changes:**

N/A

**Strengths And Weaknesses:**

Strength:
* The paper topic is interesting, and the number of hypothesis discovery surveys is still limited.
* The paper structure is clear, as presented in Figure 1.
* This work reviews enough number of works to conduct the survey. The survey covers the area of knowledge discovery.

Weakness:
* There are some types. For example, the "Multiple-Choice-Based Evaluation" has additional next line space.
* More discussion of the future work and the insights of knowledge discovery. The current survey only lists papers and introduces them in different sections. However, the discussion of future work and the insight into knowledge discovery is limited.
* For the summary, it is better to give more insight into this area.

---

> ### Author Response · Authors · 2025-07-11
> **Feedback to reviewer 1sKv**
>
> We thank Reviewer 2 for the positive feedback and constructive suggestions. We understand that your main concern is the need to add more insight into the "Discussion and Future Directions" for each section and the overall summary. We have addressed these points in our revision as follows.
>
> ### **1, Typos.**
> Thank you for catching the formatting issue. We have corrected the extra line space you pointed out and have carefully proofread the manuscript for other similar typos.
>
> ### **2, More insight in discussion, future work and summary**
> - For Hypothesis Generation: We added new insights into the evaluation of natural language hypothesis generation, highlighting the biased paradigm in the current definitions of a "good hypothesis" and "good abduction."  *(Section 4.3)*
> - For Hypothesis Discovery: We state that building a rule-learning environment requires a set of rules unknown to the model and a sufficiently expressive action space. We will provide more detailed intuition on how to achieve these requirements. *(Section 7.4)*
> - For the Summary: We added further insights, identifying the main challenges that must be tackled first and presenting a broader vision for how LLMs can be used to advance automated hypothesis discovery.  *(Section 8)*

---

### Review · Reviewer_G3US · 2025-07-05

**Summary Of Contributions:**

The paper surveys the literature of hypothesis discovery and rule learning with LLMs. The authors adopt Peirce's framework for hypothesis discovery, and organize the survey according to its stages of abduction, deduction, and induction (the corresponding sections are hypothesis generation, application, and validation). Each of the three sections review relevant methods and evaluation strategies for both formal and natural languages, then concludes with a discussion and future directions.

Section 7, Hypothesis Discovery, synthesizes the three stages into holistic hypothesis discovery, and categorizes discovery into three different types which more and more resemble real-world hypothesis discovery. In passive discovery, observations are drawn from a fixed dataset, and the LLM is passive: it doesn't have any control over which observations come next. Proactive discovery generates new observations through deduction either from the LLM's parametric memory, or by simple tools. Lastly, the LLM interacts with the environment and gathers meaningful observations through the sequence of its actions in a simulation of the real world.

**Audience:**

Yes

**Broader Impact Concerns:**

No concerns.

**Claims And Evidence:**

Yes

**Requested Changes:**

The last paragraph of 4.1.2 cites Bricken, et al., which is about mechanistic interpretability. I believe the intended citation is sparse autoencoders, which is Makhzani, A., & Frey, B. (2013). K-sparse autoencoders. arXiv preprint arXiv:1312.5663.

The summary of Chen et al. in 4.2.2 sounds very similar to program synthesis, it would help to detail and emphasize the difference.

The last paragraph of the first part of Section 5 says that there is little motivation to leverage LLMs for deductive reasoning in formal hypotheses. That's not strictly so because some work tries to improve LLM's deductive reasoning capabilites, for example, Morishita, T., Morio, G., Yamaguchi, A., & Sogawa, Y. (2023, July). Learning deductive reasoning from synthetic corpus based on formal logic. In International Conference on Machine Learning (pp. 25254-25274). PMLR.

Naming sections similarly to Figure 1 (like "Hypothesis Generation with Abduction", for example) would further emphasize that the paper is organized according to Peirce's framework.

The sentence structures in the second paragraph of 5.2 could be improved as they are very similar to each other. They start with "Similarly", "In addition", "Moreover".

In the first part of 6.2.2, the sentences starting with "Their validation process showed"  and "During evaluation" are probably in the wrong order.

Some abbreviations are not defined:
- First Order Logic (FOL)
- the IBE paradigm is defined on page 1, but then used next on page 13

Typos:
- To encourage diversity, il Lee et al. (2025)
- "Abstract Reasoning Corpus" => "Abstraction and Reasoning Corpus"

**Strengths And Weaknesses:**

## Strengths

I think that adopting Peirce's framework not only provides a clear organization for the paper, but also gives it a sound theoretical basis that helps pinpoint the gaps between real-world hypothesis discovery and current practice. For example, Section 6 discusses the one-off nature of induction and the lack of iterative updating of confidence in current practice.

I found the discussions interesting and valuable.

The paper is very clear and easy to read.

## Weaknesses

Even though the survey is about LLMs, it could reference some prior foundational work or perhaps summarize it to give some context. Currently it feels somewhat isolated from the rest of the literature and the vast non-LLM prior work.

5.1 talks about LLMs as formal language parsers. I believe that the natural language is parsed into formal language here, so the LLMs are used as natural language parsers or autoformalizers.

---

> ### Author Response · Authors · 2025-07-11
> **Feedback to Reviewer G3US (Part 1)**
>
> We sincerely thank Reviewer 1 for the thorough and constructive review. We are grateful for your positive feedback, particularly your recognition that our use of Peirce's framework provides a strong theoretical basis and clear organization. Based on your valuable suggestions, we have revised the manuscript. Our point-by-point responses are below.
>
> ### **1, Include non-LLM prior work for hypothesis generation**
> We agree that including some context on non-LLM prior work would add more insight to the paper. We have added a paragraph in the Background section to include a discussion and summary of pre-LLM hypothesis discovery works.  *(Section 2 Background)*
>
> ### **2, Term Naming**
> Indeed, "autoformalizer" is a better term to express the role of the LLM in the task of translating natural language into formal language. We have replaced "formal language parser" with "autoformalizer" in our manuscript. *(Section 5.1)*
>
> ### **3, Incorrect Citation**
> Thank you for this suggestion and for highlighting the foundational work on Sparse Autoencoders. We appreciate the opportunity to clarify our citation choice.
>
> Our survey discusses the paper "Sparse Autoencoders for Hypothesis Generation" by Movva et al. (2025). The methodology in that paper directly builds upon the modern application of SAEs pioneered by the Anthropic team (Bricken et al., 2023) to analyze the superposition problem (single LLM fired neurons may have different semantic meanings) and extract interpretable features from LLMs.
>
> Because our analysis is focused on this specific lineage of work, we believe citing the Anthropic paper is the most direct and relevant reference for the reader. While we acknowledge the crucial, foundational contributions of earlier papers like Makhzani & Frey (2013), the Anthropic paper provides the immediate methodological context for the work we are surveying.
>
> To provide a more complete picture, we have revised the manuscript to briefly acknowledge the most recent SAEs used on LLMs while retaining the more proximate Anthropic citation to best serve the reader's understanding of the specific research discussed. *(Section 4.1.2)*
>
> ### **4, Difference between formal hypothesis generation and program synthesis**
> The primary distinction between formal hypothesis generation and program synthesis is rooted in the nature of their problem specifications. Program synthesis operates on computationally concrete and self-contained problems, where the goal is to implement a well-defined pattern using abstracted data types (e.g., lists, integers). The causal relationships between inputs and outputs are explicit and are often clearly stated in the specifications.
>
> In contrast, formal hypothesis generation addresses more holistic and semantically rich inputs. The central challenge precedes implementation: one must first interpret the phenomenon to define the problem itself and abstract necessary features from information-rich input with domain knowledge and commonsense. Even in tasks with well-structured matrix inputs like the Abstraction and Reasoning Corpus (ARC), success depends on interpreting the semantic content of the input grids, such as determining which special shapes or color patterns are related to the output and how. This need for deep semantic interpretation is even more pronounced in real-world scenarios, like generating a program to analyze ecological data, where the underlying rules are not provided.
>
> Therefore, hypothesis generation involves a crucial preliminary step of semantic understanding to discover an unknown rule, a step that is already assumed to be complete in program synthesis. *(Section 4.2.2)*

---

> > ### Author Response · Authors · 2025-07-11
> > **Feedback to Reviewer G3US (Part 2)**
> >
> > ### **5, Clarification for “no motivation for formal hypothesis application”**
> > Thank you for this comment, which allows us to clarify our position. We have revised the text to better articulate our reasoning.
> >
> > While it is true that much research leverages formal language corpora to teach LLMs to reason, the focus of this survey is on how LLMs can **advance** the hypothesis discovery cycle. For the specific task of applying a formal hypothesis to gather new observations, an LLM does not offer a better solution than a dedicated formal logic solver (e.g., an FOL prover). The papers mentioned by the reviewer aim to improve the general deductive abilities of LLMs by training them to replicate the outputs of these solvers. This approach uses the solver's perfect solution as ground truth and enables the automatic generation of training data. Their goal is not to propose LLMs as a superior tool for this specific hypothesis application task.
> >
> > Based on its fundamental mechanism, a probabilistic LLM can, at best, only approach the performance of a deterministic formal solver in this closed setting; it can never be equally correct or superior. Therefore, for the purpose of advancing hypothesis discovery, there is little to be gained by using a non-transparent LLM for a task that a solver can already execute perfectly. This is the rationale for our statement. *(Section 5)*
> >
> > ### **6, Section naming**
> > Thank you for pointing out this inconsistency in our section names; we've already fixed this in our revised manuscript.
> >
> > ### **7, Sentence structure**
> > Thank you for pointing this out. We have revised the paragraph to reduce redundancy and improve its flow. *(Section 5.2)*
> >
> > ### **8, Sentence Order**
> > Thank you for the opportunity to clarify this point. The sequence is correct, but we understand the potential for confusion. The first “validation process” refers to the human annotation phase: each example was labeled by multiple annotators, and the authors show that the resulting “strengthener” and “weakener” labels are both high-quality and consistent among annotators. The phrase “during evaluation” in the next sentence describes the subsequent LLM experiment, which was conducted only after the dataset had passed these quality checks. We have rewritten the paragraph to remove this ambiguity.  *(Section 6.2.2)*
> >
> > ### **9, Abbreviations and Typos**
> > Thanks for pointing this out. We have added the definition for FOL and IBE upon their first appearance. We have also fixed the typos in our revision.

---

> > > ### Comment · Reviewer_G3US · 2025-07-14
> > > **Response to authors**
> > >
> > > Thank you for your insightful responses and your thorough modifications. All of my concerns were addressed. I believe that the revisions have resulted in an even stronger and excellent paper.

---

### Author Response · Authors · 2025-07-11
**Feedback to reviewers**

Dear reviewers,

Thank you for your constructive comments. We have responded to each point in detail and revised the manuscript accordingly.
If you have any further questions, please let us know—we will be happy to address them before the discussion period ends.

Best regards,

---

### Decision · Action_Editor_wT7K · 2025-08-12

**Recommendation:** Accept as is

**Audience:**

Yes

**Audience Explanation:**

Because discovering hypothesis and rules from large language models (LLMs) is an emerging direction that has received growing research interests, the survey will appeal to AI researchers and scholars in different fileds and offer valuable insights.

**Claims And Evidence:**

Yes

**Claims Explanation:**

This paper has presented a comprehensive survey on hypothesis discovery and rule learning with large language models (LLMs). This is an area of growing research interests. The work will appeal to AI researchers and scholars across different research fileds and offer valuable insights. The authors have surveyed diverse studies, and organized them into a coherent narrative story. All reviewers consider this paper as an outstanding contribution to survey literature.